# A Review on the Role of Endophytes and Plant Growth Promoting Rhizobacteria in Mitigating Heat Stress in Plants

**DOI:** 10.3390/microorganisms10071286

**Published:** 2022-06-24

**Authors:** Shifa Shaffique, Muhammad Aaqil Khan, Shabir Hussain Wani, Anjali Pande, Muhammad Imran, Sang-Mo Kang, Waqas Rahim, Sumera Afzal Khan, Dibya Bhatta, Eun-Hae Kwon, In-Jung Lee

**Affiliations:** 1Department of Applied Biosciences, Kyungpook National University, Daegu 41566, Korea; shifa.2021@knu.ac.kr (S.S.); aqil_bacha@yahoo.com (M.A.K.); m.imran02@yahoo.com (M.I.); kmoya@hanmail.net (S.-M.K.); divine@knu.ac.kr (D.B.); eunhaekwon@naver.com (E.-H.K.); 2Mountain Research Center for Field Crops Khudwani, Shere-e-Kashmir University of Agriculture Sciences and Technology Srinagar, Anantnag 190025, Jammu and Kashmir, India; shabirhwani@skuastkashmir.ac.in; 3Laboratory of Plant Molecular Pathology and Functional Genomics, Department of Plant Biosciences, School of Applied Biosciences, College of Agriculture and Life Science, Kyungpook National University, Daegu 41944, Korea; anjali.pande23@gmail.com (A.P.); waqasrahim999@yahoo.com (W.R.); 4Centre of Biotechnology and Microbiology, University of Peshawar, Peshawar 45000, Pakistan; drsumera@uop.edu.pk

**Keywords:** heat stress, bio stimulant, microbes

## Abstract

Among abiotic stresses, heat stress is described as one of the major limiting factors of crop growth worldwide, as high temperatures elicit a series of physiological, molecular, and biochemical cascade events that ultimately result in reduced crop yield. There is growing interest among researchers in the use of beneficial microorganisms. Intricate and highly complex interactions between plants and microbes result in the alleviation of heat stress. Plant–microbe interactions are mediated by the production of phytohormones, siderophores, gene expression, osmolytes, and volatile compounds in plants. Their interaction improves antioxidant activity and accumulation of compatible osmolytes such as proline, glycine betaine, soluble sugar, and trehalose, and enriches the nutrient status of stressed plants. Therefore, this review aims to discuss the heat response of plants and to understand the mechanisms of microbe-mediated stress alleviation on a physio-molecular basis. This review indicates that microbes have a great potential to enhance the protection of plants from heat stress and enhance plant growth and yield. Owing to the metabolic diversity of microorganisms, they can be useful in mitigating heat stress in crop plants. In this regard, microorganisms do not present new threats to ecological systems. Overall, it is expected that continued research on microbe-mediated heat stress tolerance in plants will enable this technology to be used as an ecofriendly tool for sustainable agronomy.

## 1. Introduction

Heat stress is defined as the rise in the temperature of 10–15 °C above ambient. Heat stress negatively affects plant growth and development at all stages, from germination to harvesting [1,2,3]. Plants are sessile in nature and are exposed to variable temperature ranges. The optimal temperature for plant growth is 60–75 °F [4,5]. A high temperature is an environmental hazard and leads to abiotic stress that limits crop yield. Among all abiotic stresses such as drought, salinity, heavy metal exposure, and temperature, heat stress has the most devastating effect on plant metabolism and growth. Temperatures greater than 75 °F are referred to cause heat stress. Above the normal temperature range, plants restrict growth, development, and physiological cellular metabolism. Heat stress raises the morbidity and mortality of plants and deteriorate their quality [6,7,8]. If the duration of heat stress increases, it may cause irreversible changes, such as cellular destruction, in plant cells. Plants show various signs of heat stress, such as wilting, leaf damage, fruit drop, blossom end rot, and bolting [9,10].

Higher temperatures lead to a cascade of cellular functions and the release of heat shock proteins (HSPs), which minimize cellular damage in plants. Heat stress affects the physiological processes of plant growth and development in several ways [11,12]. Several studies conducted worldwide suggest various regulators of heat stress using different omics approaches, as shown in Table 1. Heat stress increases membrane fluidity, leading to the uncoupling of a reaction series resulting in altered metabolism, and impairs cell machinery and chromatin changes in plants. The uncoupling of reactions leads to accretion of intermediate products and reactive oxygen species in plant cells. Heat stress that turns the central dogma blocks the degradation of proteins and disturbs the cytoskeleton of plant cells [13,14,15]. The thylakoid membrane of chloroplasts falls off in response to heat stress, which minimizes the function of the electron transport chain and impairs photosynthesis in the photosystem II (PSII) [16]. A comparative analysis of the gradual heat stress response and shock heat stress response was conducted in strawberry plants [17]. The results showed a high level of peroxidase and minimal protein content. Increased peroxidase activity is involved in thermotolerance [17]. *Triticum aestivum* subjected to heat stress restricted plant seedling characteristics and germination index. The plant produced reactive oxygen species and antioxidant enzymes that impaired photosynthesis and degraded proteins, thereby affecting the entire germination process [18].

Heat stress is an extremely serious issue that is responsible for extensive crop loss and will likely worsen in the future [19,20,21]. Temperatures above the optimal threshold value have a negative impact on crop physiology from mild to permanent damage. Since heat stress is a direct consequence of climate change, which ultimately increases the frequency of heatwaves, resulting in global warming [22], ensuring plant recovery and survival becomes a major challenge [23,24,25]. Moreover, as global warming worsens daily, strategies to enhance plant thermotolerance are urgently needed.

Various measures can be taken to minimize heat stress such as shading and deep-water planting of vulnerable plants. Furthermore, microbes are fundamental living components on Earth that provide sustainability to plants against various stresses and provide nutrition and resistance to combat diseases [26,27,28]. Microbial application is an advanced, globally accepted, environmentally friendly, and sustainable technique that uses soil microbes in stress-compromised plants to lessen the lethal effects of ecological stress. It is cheaper, ecofriendly, and easily available; it can be adopted and applied to produce high-quality yields. Plant–microbial interactions enhance the accessibility of plants towards organic materials [29]. They also play an important role in sustainable agronomy and ecology. Plant growth-promoting rhizobacteria synergistically improve plant growth by producing phytohormones, minimizing stress levels, and elevating the nutritional status [30,31]. Cyclic phosphorylation induces the expression of HSPs, which are molecular chaperones that may also be produced by galactinol synthases [13,32]. Beneficial microorganisms associated with plants can improve their resistance towards biotic and abiotic stresses. They alleviate adverse effects of stress and promote plant growth [33,34,35]. It is important to explore the plant microbial community that contributes towards providing resistance against different environmental stresses. Only a few studies have reported plant–microbial interactions in tackling heat stress in plants. The aim of this study was to gather information regarding plant microbial endophytic and rhizospheric communities in the mitigation of heat stress. Such information may enhance our current understanding of the role of microorganisms in plant stress mitigation and thus enabling their use in a strategy to attain sustainable agriculture under changing climatic conditions.

**Table 1 microorganisms-10-01286-t001:** Physiological and molecular responses in plants against heat stress (NA: Not Applicable).

References Country Year	Plants	Model/Approach	Heat Stress Regulators
Kotak et al. [22] 2007	*Arabidopsis*	Omics	Phytohormone HS MBF1c *HOT2*
Postgate et al. [36] 2013	*Arabidopsis*	Microarray	HSP70 HSP60 APX
Peoples et al. [37] 2007	NA	Appraisal	TATA box proximal 5′ flanking regions
Allahverdiyeva et al. [38] 2004	*Arabidopsis*, *Lycopersicon esculentum*	Experimental	HsfA1,2 HsfB1
Giller et al. [39] 2001	*Lycopersicon esculentum*, *Citrullus lanatus*	Experimental	Phenolic components
Szymanska et al. [12] 2011	NA	Appraisal	*Dhn*, *Sag*, *Sgr*
Ghosh et al. [17] 2004	*Fragaria × ananassa*	Experimental	Antioxidant enzymes
Saha et al. [18] 2010	*Triticum aestivum*	Experimental	Cellular, molecular and metabolic cascade
Glick et al. [40] 1999	*Soybean and Arabidopsis*	In vivo and In vitro	HSP90 HSP60 HSP20

## 2. Role of Microorganisms in Thermotolerance

Microbes are biological control agents that combat heat stress. Microbial inoculation causes thermotolerance. Exopolysaccharides are released by bacteria under heat stress containing 97% water, which improve the soil structure. Water remains available to plants, so it is helpful during the stress period. Thermotolerance is a complex mechanism. It has also been hypothesized that the production of proline and glycine betaine contributes to thermoregulation [41,42]. The details of the two major categories of thermotolerant microbes are discussed as below.

### 2.1. Endophytes

Endophytes are microorganisms that live in plant cells and form biofilms that interact with plant exudates. Endophytes are used as biostimulants to produce various compounds. Most endophytes are inaccessible because they live inside plant tissues and remain in symbiotic relationships [43,44]. In the light of plant–microbe interaction-mediated heat stress mitigation, only limited studies are available. Endophytes form symbiotic relationships with plants. To maintain a stable relationship, they produce various kinds of compounds that promote the growth and development of the plants. They produce biochemicals that cannot be synthesized [45,46]. Evidence of endophytes, their mode of action, and their growth-promoting traits have been reported by increasing number of recent publications, indicating their importance.

A field experimental study was designed to evaluate the inoculation effects of endophytic microbe SA187 on *Arabidopsis thaliana* and wheat plants [47]. The plants were divided into two groups: the untreated normal group, and the group inoculated with *Enterobacter* SA187. The plants were exposed to high temperatures up to 44 °C to induce heat stress. Agronomic traits were also assessed. *Enterobacter* sp. SA187 induced thermotolerance in plants and promoted thermopriming. The results were repeated over three consecutive seasons. The inoculation treatment group showed an increase in overall plant biomass and height by 10–14%, grain yield by 40%, and seed weight by 12%. These results suggest that SA187 inoculation is beneficial to plants to enhance the heat tolerance [47].

Meena et al. studied tomato plant seedlings subjected to heat and drought stress. *Septoglomus deserticola* and *S. constrictu* were inoculated, and cellular parameters were measured. Inoculation decreased oxidative stress and minimized the level of reactive oxygen species. The symbiotic effect improved cellular performance, stomatal conductance, and leaf water content. The mycorrhizal inoculation improved and enhanced physiological features under combined stress [48].

The inoculation of *B. cereus* SA1 on soybean plants under heat stress conditions causes thermoregulation [49]. The analysis showed increased chlorophyll a and b, carotenoid, protein, ascorbic acid peroxidase, and superoxide dismutase levels in plants. SA1 significantly improved thermoregulation [49] (Table 2).

### 2.2. Plant Growth-Promoting Rhizobacteria (PGPR)

Bacteria that colonize in the roots of plants or along the rhizopheric axis and promote plant growth are known as PGPR. They promote plant growth directly by regulating nutritional status (phosphate solubilization, N fixation, iron sequestration) and hormone synthesis (IAA, GB, CK, etc.). They also promote plant growth indirectly by providing immunity to plants against environmental stress and by producing compatible solutes such as proline, sugars, organic acids, and glycine betaine [50,51]. Table 3 summarizes the recent studies on inoculation of PGPR to plants under heat stress, their effects, and mechanism of action. Overall, the inoculation of PGPR is beneficial to the plants to overcome the deleterious effects of heat stress as illustrated in Figure 1.

Microbes are beneficial for the thermotolerance of plants. In the field experiments, *Triticum aestivum* was selected as a model plant for inoculation with *Bacillus amyloliquefaciens* UCMB5113 and was exposed to short-term heat stress. Glutathione reductase and transcription factors were selected as gold standards for comparative analysis. The results showed that the inoculated plants had reduced APX1, GR, SAMS1, and HSP17 expression. The recovery and survival of inoculated plants were more significant than those of non-inoculated plants [52]. The inoculation of *B. cereus* SA1 on soybean plants under heat stress conditions caused thermoregulation. The analysis showed increased chlorophyll a and b, carotenoid, protein, stress tolerant enzymes levels in plants. SA1 significantly improved thermoregulation [49].

In 2021, in vitro experimental plants grown in a growth chamber suggested that PGPR play a key role in thermotolerance. The bacterial species *Bacillus cereus*, *Pseudomonas* spp., *Serratia liquefaciens*, *P. fluorescens*, and *Pseudomonas putida*, which were hosted in *Solanum lycopersicum* L, *Cajanus cajan*, *G. max*, and *Triticum* spp., caused the production of phytohormones, antioxidant enzymes, and ACC-deaminase consequently mitigated heat stress [53].

The microbial isolates *Burkholderia phytofirmans* PsJN, *Curvularia proturberata* isolate Cp4666D, which were hosted in *T. aestivum*, *Dichanthelium lanuginosum*, and *S. lycopersicum* caused remarkable thermotolerance in plants. Microbes increased the production of IAA and cytokines, as well as the molecular protein and chlorophyll contents, suppressing plant pathogens and production of free radicals [54].

Kang et al. conducted an experimental study in the Republic of South Korea on *Glycine max* (Soybeans) that was subjected to heat stress to reveal the effect of microbial inoculation. The heat stress conditions were a day/night cycle of 16 h at 38 °C and 8 h at 30 °C for 1 week. Soybeans are highly sensitive to heat stress, and the use of PGPR counters the negative effects of heat stress. The bacterial strain *Bacillus tequilensis* SSB07 was inoculated into the plant and various growth attributes such as seedling growth and production of GBs, IAA, and ABA were recorded. SSB07 increased shoot length and biomass. SSB07 countered the negative effects of heat stress on crop growth and development [55].

The *Pseudomonas* sp. strain AKM-P6 was inoculated into sorghum. Sterilized seeds were smeared with a talc-based formulation (108 cells/g) of strain AKM-P6 and sown in plastic cups. Five-day-old seedlings were exposed to heat stress and harvested after 10 d. The samples were analyzed using a scanning electron microscope, and other plant biochemical parameters were assessed. The conditions were a temperature of 47–50 °C during the day and 30–33 °C at night for 10 days. The microbial strain AKMP6 improved the high temperature stress in sorghum seedlings. The microbial strain AKMP6 helped sorghum seedlings endure and grow at preeminent temperatures for up to 15 days, whereas the inoculated plants died after 5 days. Bacterial inoculation promoted the biosynthesis of high-molecular-weight proteins in leaves under increased temperature, compact membrane injury, and increase in the levels of cellular metabolites. The strain AKM-P6 augmented the lenience of sorghum seedlings by inducing physiological and biochemical changes in the plants [56].

*Triticum aestivum* was inoculated with *Pseudomonas putida* strain AKMP7. The disinfected seeds were planted in plastic pots. After 2 weeks, each seedling was dispersed in one pot. The plants were unprotected from heat stress, and after 95 d, the seedlings were collected to assess their growth and enzymatic activities on the 110th day of growth. The temperature conditions were 37–40 °C during the day and 27–30 °C at night for 95 d. The inoculation of AKMP7 increased the levels of cellular constituents, plant development, and total biomass. AKMP7 also convalesced with the survival and growth of wheat plants under heat stress by increasing their root and shoot length, dry biomass, tiller, spike, grain formation, and reducing membrane injury and antioxidant enzyme activities such as SOD, APX, and CAT activities under heat stress. The results showed that AKMP7 could be effective in relieving heat stress and subsequently improving the growth of wheat plants under heat stress [55]. In other trials, the thermotolerance potential of *G**lycine max* inoculated with *Bacillus aryabhattai* SRB02 and subjected to 38 °C/30 °C day/night heat stress for 0, 12, and 48 h was measured. At vegetative stage 3 (V3), 10 mL of bacterial culture (1 × 10^8^ cfu/mL) was applied for 3 days. Growth parameters were recorded after the application of heat stress. SRB02-treated soybean plants showed significantly better thermotolerance than the untreated plants, based on their ABA-mediated stomatal closure and increased IAA, JA, and GA contents, plant growth, and biomass. SRB02 also endured extraordinary nitrosative stress induced by the nitric oxide donors GSNO and CysNO. These results suggest that SRB02 may be a valuable source of biofertilizers to increase crop production [47].

In 2014, Abd El-Daim performed an experimental study on microbial mitigation of heat stress in wheat crops. *Triticum aestivum* seeds were soaked in a bacterial suspension (1 × 10^7^ cfu/mL) for 2 h at 28 °C and grown in pots in a growth chamber for 12 days. The plants were subjected to a higher temperature of 45 °C, following which the expression levels of ascorbate peroxidase (APX1), S-adenosylmethionine synthetase (SAMS1), HSP17.8, heat-inducible transcription factor (HsfB1), heat shock factor 3 (HsfA3), and MBF1c was determined. Bacterial treatment improved the heat stress control in both cultivars of wheat by levering the transcription levels of several stress-related genes and ascorbate-glutathione enzymes. Seeds treated with two PGPR strains that colonized the roots amended their thermotolerance [57].

In 2015, another scientist, Meena, conducted an experiment using another bacterial species to observe the effect of beneficial microbes under heat stress in plants. A T. aestivum cultivar (HUW-234) was inoculated with *Pseudomonas aeruginosa* strain 2CpS1 in a greenhouse experiment. The growth chamber conditions were 30 °C/25 °C day/night for 24 days. Strain 2CpS1 increased the seedling length, leaf area, biomass, total chlorophyll content, relative water content, and soil moisture contents. The application of stress-tolerant PGPR strains could be used as a reasonable approach for cultivating crops at elevated temperatures [48].

In another experiment, *T. aestivum* were subjected to high temperature stress, and their thermotolerance potential towards heat stress was recorded. Seeds were grown in hydroponics for 7 days, and after one day of microbial constrain of *SN13* inoculation, seedlings were exposed to 45 °C. After completing the experiment, physiological and biochemical factors such as membrane potential, osmolytes accumulation, proline content, lipid peroxidation, total soluble sugar, and six stress-responsive genes were assessed. The results suggest that SN13 positively controls the expression of stress-responsive genes and phytohormones, suggesting its multidimensional role in stress response. The differential responses of rice seedlings to heat stress and phytohormones were confirmed using principal component analysis (PCA) based on the effects of SN13 inoculation on the response of rice to heat stress and phytohormone treatments [58].

Moreover, 23-day-old sprouts of *Lycopersicon esculentum* were bio-primed with a bacterial inoculum of *Paraburkholderia phytofirmans* strain PsJN at 10^6^ CFU/mL and planted into the green house. Leaf gas exchange ratio, chlorophyll fluorescence rate, photosynthetic pigment content, and other parameters were evaluated. The greenhouse temperature was maintained at 32 °C under 16 h of light and at 27 °C for 8 h of dark for 45 days. PsJN improved plant growth attributes such as chlorophyll content, photosystem II, and sugar and total protein content. The PsJN strain can improve the destructive effects of heat stress by stimulating the thermotolerance mechanism of tomato plants [52].

**Table 3 microorganisms-10-01286-t003:** Application of plant growth-promoting rhizobacteria in mitigating heat stress (↑: Increase Traits, ↓: Decrease Traits).

References Country Year	Microbes	Model	Plant	Parameters	MOA	Stress	Effect
Abd El-Daim et al. [57] 2014	*Bacillus amyloliquefaciens* UCMB5113	Field	*Triticum aestivum*	↑ Survival * rate	↓ GR ↓ APX ↓ HPS17	Short	Beneficial
Rana et al. [54] 2012	*Curvularia proturberata* isolate Cp4666D, *Burkholderia phytofirmans* PsJN	Field experiment	*Triticum aestivum*, *Dichanthelium lanuginosum*, *Solanum lycopersicum*	Production of IAA, cytokines, protein and ↑ chlorophyll	↓ Pathogen, ↓ ROS	Heat stress	Beneficial
Mitra et al. [59] 2021	*Bacillus cereus,* *Pseudomonas,* *Serratia liquefaciens*, *P. fluorescens* and *Pseudomonas putida*	In vitro	*S. lycopersicum* L., *Cajanus cajan,* *G. max, and Triticum* spp.	↑ ACC-deaminase, Production, ↑ phytohormone and ↑ antioxidant defense	Thermal tolerance	High temp	Sustainable
Maitra we al. [58] 2011	*Aeromonas hydrophilla* *Serratia liquefaciens* *Serratia proteamaculans*	In vitro	*Glycine max*	↑ Exopolysacchrides production	Thermotolerance	High temp	Remarkable
Kang et al. [53] 2019	*Bacillus tequilensis* SSB07	Experimental	*Glycine max*	↑ Gibberellins ↑ IAA and ↑ ABA, jasmonic acid and salicylicacid contents	Thermotolerant	Moderate	Improvement
Ali et al. [55] 2011	*Pseudomonas putida*AKMP7	Experimental	*Triticum aestivum*	↑ Root and shoot length, ↑ biomass, ↑ SOD, ↑ CAT and APX	Thermotolerant	High	Improvement
Ali et al. [56] 2009	*Pseudomonas* AKM-P6	Experimental	Sorghum	↑ Cellular metabolites	Thermotolerant	High	Improvement
Park et al. [47] 2017	*Bacillus aryabhattai* SRB02	Experimental	*Glycine max*	↑ ABA ↑ IAA, JA, GAs contents,	Fertilizers+ thermotolerance	Medium	Improvement
Meena et al. [48] 2015	*Pseudomonas aeruginosa* 2CpS1	Net house experiment	*Triticum aestivum cultivar* (HUW-234)	↑ Plant height and root length, ↑ chlorophyll content,	Mitigation	High	Improvement
2018	*Bacillus amyloliquefaciens*	Experimental	*Oryza sativa*	↑ Proline, Total Soluble Sugar, ↑ Lipid Peroxidation and over expression of six stress-responsive of dehydrin (DHN), glutathione S- protein 6 (NRAMP6) genes	↑ Modulated stress-responsive gene expressions ↑ phytohormone	High	Significant
Issa et al. [52] 2018	*Paraburkholderia* *phytofirmans* PsJN	Green house experiment	*Lycopersicon esculentum*	↑ Growth Biomass Chlorophyll content	↑ Chlorophyll content, Photosystem II, ↑ Accumulations of sugars, total amino acids, proline, and Malate.	Thermotolerant	Improvement

## 3. Physiological Changes Induced by Thermotolerant Microbes in Plants under Heat Stress

### 3.1. Photosynthesis

Photosynthesis is a natural cellular respiration process by which plants convert light energy into chemical energy [60]. Heat stress disrupts the photosynthetic apparatus, resulting in the inhibition of plant growth and development. Studies have suggested that it inhibits the production of ribulose 1, 5-bisphosphate (RuBP), which is involved in the electron transport chains [61,62]. Heat stress also inactivates enzymes involved in photosystem 11 lowering the rate of photosynthesis [63,64]. However, under heat stress, oxygenic microbes contain light-harvesting pigments that induce the reprogramming of cellular events in the thylakoid membrane [65,66,67]. They release oxygen and absorb carbon dioxide. Cyanobacteria are one of the bacteria that support photosynthesis and promote plant growth [68].

### 3.2. Changes in Respiration

Respiration is a chemical process that involves oxygen and glucose to produce energy for plant survival and is important in maintaining plant growth as well as the carbon cycle [69,70]. Higher temperatures enhance cellular respiration owing to the increased kinetic energy. However, most of this energy is apportioned to maintain respiration, resulting in a general reduction in the energy utilization efficiency of plants [70,71]. Beneficial microbes increase soil respiration and improve nutrient cycling in plants. Microbes can minimize stress levels and restore ecosystems to an equilibrium state. Plant–microbe interactions maintain nitrogen, hydrogen, sulfur, and oxygen levels in a biogeochemical cycle [72,73].

### 3.3. Stomatal Closure

Stomata are microscopic openings present in the epidermis of leaves. Stomatal opening and closing are important for maintaining the physiological functions such as transpiration, and stomatal closure is a common adaptative response to heat stress [74,75]. Under heat stress, there is a possibility of rapid water loss. Plant–microbial interaction enhances the production of abscisic acid (ABA), a phytohormone, also known as a stress hormone, that activates various biotic and abiotic stress conditions, causes the closure of stomata, and is also important in osmoregulation [76,77]. The microbial production of ABA causes simultaneous stomatal closure [78,79]. Plants close their stomata to reserve water loss caused by evaporation [75].

## 4. Molecular Mechanism of Action of Microbes in Mitigating Heat Stress in Plants

### 4.1. Nitrogen Fixation

Nitrogen fixation is a natural process by which gaseous N_2_ is converted into biological forms of NH_3_ and NH_4_. Nitrogen is a macronutrient in plants. Heat stress promotes N accumulation in the meristems of plant cells via apical blade erosion and plays a vital role in energy metabolism, protein synthesis, and photosynthesis [80,81]. Higher temperatures delay the development of nitrogenase activity in plants, resulting in inhibition of N fixation, leading to stunted growth of plants [82,83]. Microbes can mitigate heat stress by enhancing N fixation. Microbes can transfer inert atmospheric N into the most reactive forms of ammonia, nitrates, and nitrites through a series of chemical reactions [84,85,86]. Thus, microbes possess a relatively symbiont relationship with plant species known as diazotrophs [87,88]. There are two kinds of nitrogen fixating microbes (symbiotic and no symbiotic) that improve the soil nitrogen concentration, rhizobacterial population levels, soil nitrogenase activities and N uptake in plants [89]. Zhang et al. in [90] tested the beneficial effect of several bacterial strain on soybean growth under suboptimal temperature and found that the bacterial growth promoting effects are caused by the bacterial nitrogen fixing potential. Various recent studies are reported in favor of mitigation of heat stress by nitrogen fixing bacteria [91,92].

### 4.2. Microbial Production of Siderophore

Siderophores are organic compounds with low molecular weights. These compounds have a high affinity for iron-chelating compounds. Microbes can produce siderophores, which are microscopic, high-affinity iron-chelating combinations. These serve primarily to transport iron across the cell membranes through membrane receptors. Various gram-positive and gram-negative bacteria produce and secrete siderophore to scavenge iron from the environment [93,94]. Plant–microbial interactions enhance siderophore production, ultimately improving the nutritional status and growth of plants under stress [95]. Application of siderophore producing bacteria Pseudomonas putida and Pseudomonas sp. have shown the improvement in growth, chlorophyll and plant biomass in wheat and sorghum under heat stress [55,56].

### 4.3. Microbial Production of 1-Aminocyclopropane-1-carboxylate (ACC) Deaminase

Some microorganisms can produce the enzyme ACC-deaminase and promote plant growth by sequestering and splitting plant-produced ACC, producing α-ketobutyrate and ammonia, which lowers the level of ethylene in plants. This is the most efficient mechanism of action for plants to tolerate stress and promotes a much easier lifestyle in the soil [96,97,98]. Plant–microbial interactions enhance ACC-deaminase production, which facilitates plant growth under stress conditions [99]. Recently reported ACC deaminase activity produced Achromobacter piechaudii, which moderated ethylene metabolism and ultimately resulted in better heat tolerance in pepper [100]. Furthermore, ACC deaminase producing Brevibacterium linens enhance combined heat and UV-B radiation stress in rice plant and enhance plant biomass, photosynthetic traits and decrease ethylene emission [101]. In another study of Mukhtar et al. in [102], they reported that ACC deaminase producing Bacillus cereus mitigate heat stress in tomato and observed drastic morphological and physiological effects on tomato plants under heat stress.

### 4.4. Microbial Production of Phytohormone

In response to stress, microbes can produce phytohormones that act as endogenous growth regulators by reducing stress and optimizing plant growth. Microbes produce various hormones such as gibberellin (GB), cytokinin (CK), salicylic acid (SA), indole-3-acetic acid (IAA), and ABA [103,104]. The mechanism of stress mitigation involves modulating antioxidant potential and maintaining the osmolyte potential of plants. Most prominently, ABA is produced under stress conditions and is termed a stress hormone. It causes stomatal closure, preventing osmolyte loss through evaporation [57,105]. Phytohormones form signaling networks. Various studies have suggested that the exogenous application of ABA mitigates heat stress and its consequences. ABA is a vital hormone that reduces oxidative stress by activating the defense system, leading towards redox homeostasis [54,55,59]. Plant–microbial interaction also enhances the production of the GB hormone, which is important in regulation of developmental process such as germination, flowering, fruit, and leaf senescence. It controls major aspects of plant growth. Under plant–microbial interaction, there is an increase in the level of auxin, which positively modulates the genetic expression and enhances the activity of the defensive antioxidant system of the plants [106]. Auxin producing *Azospirillum brasilense* was reported to mitigate heat stress in wheat by maintaining water status [107]. Khan et al. had demonstrated how thermotolerant *Bacillus cereus* mitigate heat stress in tomato and soybean through moderation in the auxin levels. Similarly, gibberellins is another phytohormone that are produce by bacteria and are involve in all plant growth and development including stem elongation, leaf expansion and fruit ripening [108]. Atzorn et al. in [109] reported gibberellin producing bacteria first time in *Rhizobium meliloti*. Nowadays, several genera of *Pseudomonas*, *Serratia*, *Bacillus*, and *Arthrobacter bacteria* have been reported for the production of different GAs.

### 4.5. Molecular Approaches

To sustain crop yield through thermotolerance, plants evolve through a series of cascade events. Previous electronic data on heat stress phenomena are available, including data on the production of classical chaperone proteins. In Germany, an experiment was designed to evaluate the thermotolerance mechanism of *Arabidopsis*. Results showed that phytohormones (ABA, SA, and ethylene), oxidative stress, and several mutants were involved in thermotolerance. An applied molecular dynamics study revealed that the genes MULTIPROTEIN BRIDGING FACTOR 1c (MBF1c) and *HOT*2, which encode chitinase-like proteins, are involved in acquiring hemostatic heat tolerance [110].

Molecular dynamics studies have extended our knowledge of plant thermotolerance and its associated genes. Microarrays suggest that heat stress causes the development of various HSPs. Among them, HSP70, APX, and HSP60 are potent and thermodynamically involved [36]. The responses of plants to heat stress are unique. Various studies have strengthened our understanding of the vital roles of HSPs. HSP70, HSP90, and HSP20b function to analyze mutants, denature proteins, and for homooligomerization, respectively [40]. Heat stress deranges chromatin organization in plant cells. JUMONJI (JMJ proteins are responsible for chromatin organization. These are histone demethylase proteins found in nature [111,112,113]. Demethylases are enzymes that eliminate methyl groups from molecules and promote structural support to chromosomes. Plants maintain heat memory stress because of lowered H3K27me3 (histone H3 lysine 27 trimethylation) expression in small heat shock genes [114,115,116]. These are an important family of proteins that control heat shock genes, thus allow plant cells to tolerate heat stress via a memory mechanism [117,118,119].

The production and accumulation of HSPs range from molecular mass 10 kDa to 100 kDa. Studies have shown that the application of genetic stock, where heat shock elements are present in a TATA box in the proximal 5 flanking regions of heat shock genes, and the application of osmoprotectants can combat heat stress [37]. In 2011, a study on heat stress reported detailed information about the heat mechanism wherein senescence-associated genes (*sag*), dehydrins (*dhn*), and HSP stay-green (*sgr*) genes are involved in stabilizing heat stress. Plant thermotolerance can be improved through molecular breeding [12]. To minimize heat stress, more than 20 heat shock factors are involved in initiating a cascade event series to trigger, maintain, and recover the plants. Two main factors HsfA1, 2 and HsfB1 are responsible for thermotolerance. HsfB1 is a co-regulator that activates Hsf, a working horse that is activated in the summer [11]. Acute and chronic heat stress affects plant growth, development, and yield [120].

### 4.6. Microbial Production of Volatile Compounds

Plant–microbial interactions enhance the production of volatile compounds. Various microorganisms, such as Bacillus and Pseudomonas produce volatile compounds when inoculated into the host plants. These are important compounds with a range of more than 200 types such as isoprenoids and terpenoids, which are known as growth inducers. They also promote the defense system of plants. In an experimental setup of tomato and watermelon plants grown in a growth chamber for 30 days at various temperatures, the authors evaluated the responsive components against heat stress. Moreover. HSPs and other plant components are also involved in thermoregulation. The acclimatization of plants is because of the bioactivity of phenolic compounds and the inhibition of oxidation [39,121,122].

### 4.7. Organic Acid Production

Organic acid are considered as an essential source of carbon and are rich in vitality. Plant growth promoting bacteria synthesize various secondary metabolites such as phytohormones and organic osmolytes that activate host plants stress management mechanisms and solubilize nutrients for easy absorption by plants [123]. Organic acid producing bacteria alleviate heat stress and enhance plant growth in soybean and tomato [105,124,125]. Table 1 summarizes the physiological and molecular approaches of microbes in mitigation of heat stress.

## 5. Conclusions and Future Prospective

It is evident that there is an increasing demand for sustainable agronomy to meet the needs of the growing population and enhance agronomic yield without disturbing ecological components. Since 2000, there has been an increase in the demand for bio stimulants as growth promoters, and research interest and publication are increasing in the area of heat stress management using beneficial microbes [126,127]. Microorganism inoculation enhances resistance and tolerance towards heat stress. The isolation and identification of beneficial microbes in the cross-protection of plants would be highly valuable. Beneficial microorganisms alleviate the adverse effects of stress through the production of phytohormones and certain metabolites and enhance plant defense systems as shown in Figure 2 (Both images were created using Biorender.com). Plant–microbial interactions activate the antioxidant defense system. They have also been found to improve ion homeostasis by maintaining osmo-protectant levels. Therefore, it is important to explore plant-associated microbial communities. Only a few studies have reported plant–microbial interactions in heat stress regulation. Overall, this review suggests that use of microorganisms needs to be thoroughly explored in the field of agronomy for mitigating heat stress and attaining the goal of sustainable agriculture.

## Figures and Tables

**Figure 1 microorganisms-10-01286-f001:**
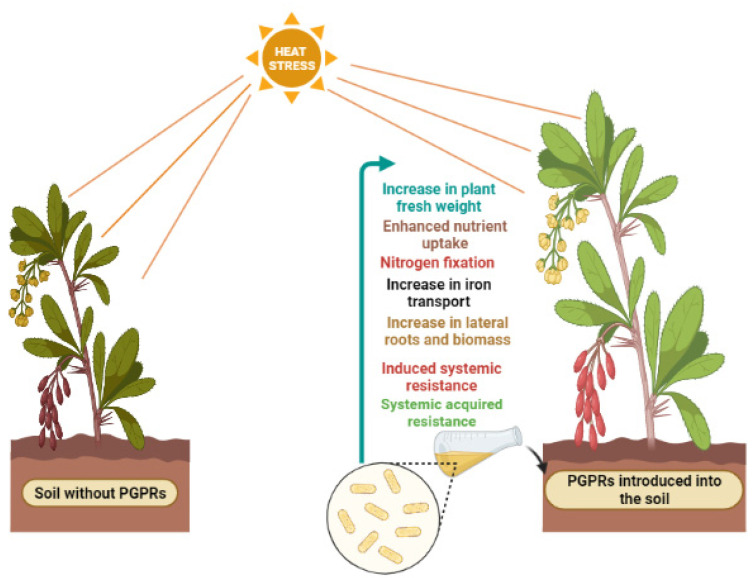
Metabolic reprogramming of the plant cell under heat stress.

**Figure 2 microorganisms-10-01286-f002:**
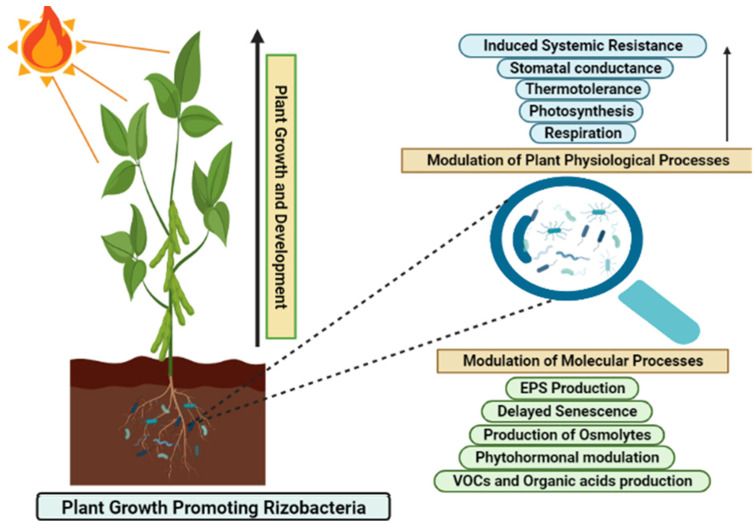
The illustration represents the role of plant growth-promoting rhizobacteria in mitigating heat stress in plants. PGPRs promote plant growth and development by modulating the physiological and molecular processes in plants.

**Table 2 microorganisms-10-01286-t002:** Application of endophytes in mitigating heat stress in plants (↑: Increase Traits, ↓: Decrease Traits).

References Country Year	Microbes	Model	Plant	Parameters	MOA	Stress	Effect
Park et al. [47] 2021	*Enterobacter* SA187	Vitro Experimental field	*Arabidopsis thaliana,* Wheat plant	↑ Biomass, ↑ 10–14% height,↑ 40% grain yield and seed weight 12%.	Chromatin modification	Long term	Beneficial
Meena et al. [48] 2018	*Septoglomus deserticola* and *Septoglomus constrictu*	In vitro	*Solanum lycopersicum*	Improved Stomal conductance, water content and leaf water content	↓ Oxidative stress	Heat +drought	Improved
Anli et al. [41] 2011	*Pseudomonas fluorescens, Pantoea agglomerans, Mycobacterium* sp., *Bacillus amyloliquefaciens, Pantoea agglomerans*	Appraisail	*Triticum aestivum*	HSP90 Antioxidant enzymes, HSTP	Thermoregulation	High temp	Significant
Anli et al. [41] 2011	*B. phytofirmans*	*NA*	*Solanum tuberosum*	↑ Proline and glycine betaine	Thermotolerance	High temperature	Good biocontrol
Bisht et al. [49] 2020	*B. cereus* SA1	Experimental	*Soybean*	↑ Chlorophyll a and b ↑ Carotenoid, ↑ Chlorophyll florescence	Thermotolerant	Medium to high temp	Bio fertilizer

## Data Availability

Data supporting the present study are available from the corresponding author upon request.

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
