# Peer review of "A Review on the Role of Endophytes and Plant Growth Promoting Rhizobacteria in Mitigating Heat Stress in Plants"

_microorganisms, 2022, doi:10.3390/microorganisms10071286_

Round 1
Reviewer 1 Report
1- Abstract is too weak and needs improvement
2- First line in the introduction "A higher temperature is an environmental hazard that limits the crop yield". There are many papers demonstrated the negative impacts of temperature on crop yield as:
- Ali, M.G.M., Ahmed, M., Ibrahim, M.M. et al. Optimizing sowing window, cultivar choice, and plant density to boost maize yield under RCP8.5 climate scenario of CMIP5. Int J Biometeorol (2022). https://doi.org/10.1007/s00484-022-02253-x
- https://doi.org/10.1016/j.agwat.2020.106626
- https://doi.org/10.1007/s42106-022-00190-8
Add these references to the first line of the introduction after highlighting the issue
3- Unfortunately, I did not find line numbers, causing difficulties in the review
4- The review neglects the method of survey and data collection!
5- The number of references throughout the text is too small, please increase it wherever.
Author Response
thankyou so much for your valuable comments.please find the attached file

Reviewer 2 Report
The paper entitled "Research progress in sustainable strategies for combating heat stress by glycocalyx microbial bio stimulant" submitted by Shifa Shaffique et al is focused on the plants' heat-triggered response and the underlying coping physio-molecular mechanisms of microbe-mediated stress alleviation.
In general, the paper is structured and written at a relatively basic level and several corrections are needed for a new submission. The overall appearance of the submitted material expresses negligence and disregard towards the MDPI Manuscript preparation rules/criteria, while Microorganisms is a Q 1 scientific journal, IF 4.128.
- Phrasing and English syntax need revision.
page 3 "Fixed nitrogen is important because they give us valuable components, such as proteins, amino acids, nucleic acids, etc. [44,45]."
2.Several misspellings, absent capital letters and phrases without verbs can be found in the whole text, even in the authors' affiliations, tables
eg in the abstract body text: "The improved antioxidant activity and accumulation of compatible osmolytes, such as proline, glycine-betaine, soluble sugar, and trehalose, enriched the nutrient status of plants"
page 2 - JUMONJI proteins (definition?) are important ? which control the ....
table 1 "Invivo and invitro", "japan"
page 9 conclusion section: "Plant microbial interaction pocess the potentia.."
3. basic level, for eg. the paragraph:
"Global warming is the greatest challenge for the agriculture industry. Plant recovery and survival are important, as global warming worsens every day; thus, developing some strategies that yield plant thermotolerance is needed. When heat stress rises, it causes microbes to acclimatize in the soil and become resilient when the temperature becomes favorable [16]."
4. several acronyms do not have any description or definition (PSII, IPCC, etc)
5. several sections and paragraphs are produced using copy - paste (pages 6-7)
Several sections - pages 9-10 are not covered (Author Contributions, Funding, Institutional Review Board Statement, etc)
Author Response
please find the attached paper

Reviewer 3 Report
The submitted manuscript is interesting.The manuscript may be helpful for scientists in biological sciences and may be of considerable interest to researchers.In my opinion, the article is good, the information presented in the tables is valuable.
Minor errors that require correction are:
- Thoroughly checking the Literature chapter.Now is disordered, namely some journal names are written in lowercase. Also add doi.
- 2) Make sure all items are cited in the text and in the table of contents, and vice versa.
- Complete the information in Author Contributions and Funding.
- Read carefully: Institutional Review Board Statement, Informed Consent Statemen, Data Availability Statemen.They should only be if they concern you.
Author Response
please find the attached comments

Round 2
Reviewer 2 Report
Please check the replies included with the cover letter.

Author Response
|
Dear reviewer, Thank you so much for your valuable comments to improve my manuscript. I have considered your suggestions and improved the content, English language mistakes, phrases, author’s acronyms, and vocabulary. I also modified the abstract and checked capital letters carefully. After careful update of all the mistakes mentioned and considering the comments of other reviewers, I finalized the paper. After that, I sent the paper for English editing for a further clarification. The English editing Certificate is also attached. I have changed the entire contents of the manuscript according to your and other reviewers’ valuable construction guidelines. I really appreciate your efforts for giving suggestions to make my manuscript capable of acceptance in a reputable journal. Thank you |
